# Diagnostic Efficacy of Bone SPECT Techniques in Differentiating Unilateral and Bilateral Condylar Hyperplasia

**DOI:** 10.3390/diagnostics14222548

**Published:** 2024-11-14

**Authors:** Luz Kelly Anzola, Natalia Venegas, Maria Clara Jaramillo, Sergio Moreno, Mauricio Hinojosa, Enrique Amador, Martin Orozco, Fernando Mut

**Affiliations:** 1Nuclear Medicine Unit, Clinica Reina Sofia, Bogota 110131, Colombia; 2Nuclear Medicine Postgraduate Program, Fundación Universitaria Sanitas, Bogota 110931, Colombia; nataliavenegasro@gmail.com; 3Maxillofacial Surgery Clinica Reina Sofia, Bogota 110131, Colombia; mcj116@outlook.com (M.C.J.); eamador15@gmail.com (E.A.); 4Clinical Epidemiologist, Universidad Nacional de Colombia, Bogota 111311, Colombia; smmorenol@unal.edu.co; 5Medicina Nuclear Sanitas Palermo, Bogota 111311, Colombia; mauriciohinojosa1405@gmail.com; 6Maxillofacial Department, Clinica Universitaria Colombia, Bogota 110911, Colombia; martinorozcofernandez@gmail.com; 7Nuclear Medicine Unit, Hospital Italiano, Montevideo 11600, Uruguay; mut.fer@gmail.com

**Keywords:** mandibular condyle, temporomandibular joint, single-photon emission computed tomography, scintigraphy, radioisotope diagnostic

## Abstract

**Objectives:** This analytical cross-sectional study evaluates diagnostic approaches for active condylar hyperplasia using bone SPECT techniques. **Methods**: it was compared the effectiveness of relative activity assessments between condyles and quantitative analysis using the condyle/clivus ratio. **Results**: This study’s findings reveal that the condyle/clivus ratio method significantly outperforms the relative uptake method, achieving a sensitivity of 90.1% (95% CI: 84.1–94) compared to 40.7% (95% CI: 33.5–48.2), and a specificity of 77.1% (95% CI: 67.4–85) versus 72.9% (95% CI: 62.9–81.5). The condyle/clivus ratio also showed a more favorable negative likelihood ratio of 0.13 compared to 0.82 for relative uptake, and a higher area under the curve (AUC) of 0.8360 versus 0.5679. Statistically significant differences were noted (*p* = 0.0001). The condyle/clivus ratio method effectively identifies affected condyles in unilateral and bilateral condylar hyperplasia cases. **Conclusions:** This study emphasizes the importance of incorporating comprehensive clinical evaluations and imaging modalities for assessing condylar growth activity, underscoring the need for tailored reference values in different populations to ensure reliable diagnostic interpretations.

## 1. Introduction

Unilateral condylar hyperplasia (UCH) is a pathological condition of the condylar cartilage of the mandible, caused by a somatic mutation resulting in the excessive effects of growth hormones (growth factors like IGF-1), which promotes temporomandibular cartilage overgrowth by enhancing chondrocyte proliferation via the mitogen-activated protein kinase/extracellular signal-regulated kinase (MAPAK-ERK) pathway [1,2,3], leading to pronounced facial asymmetry and functional impairment, such as malocclusion and altered temporomandibular joint (TMJ) function. Four observations have supported the hypothesis that UCH is primarily a disorder of the condylar cartilage: (a) the deformity of the condylar process is more extensive than that of other parts of the mandible; (b) when the temporomandibular cartilage overgrowth is present, the bone scan shows localized abnormal uptake in the condyle; (c) resection of the condylar cartilage suppresses the abnormal growth; and (d) histopathologic changes are limited to the condylar cartilage and the adjacent subchondral bone [4]. Furthermore, genetic factors may predispose individuals to this condition, emphasizing the need for comprehensive genetic studies to uncover potential biomarkers. Most patients present symptoms during adolescence or young adulthood [5], mostly between 11 and 30 years of age, and males and females are equally affected; early diagnosis and intervention are critical to prevent complications, including temporomandibular disorders, persistent pain, and permanent skeletal deformities.

Clinical evaluations, combined with advanced imaging techniques, play a pivotal role in diagnosis and treatment planning for UCH. Computed tomography (CT) is recognized as the standard imaging modality, providing detailed anatomical and morphological information essential for distinguishing condylar hyperplasia from other conditions, such as osteochondroma and malignant tumors [6]. For diagnostic purposes, cone-beam CT is often preferred due to its superior resolution and reduced radiation exposure, making it particularly useful in young patients [7]. While magnetic resonance imaging (MRI) provides excellent soft tissue characterization, its effectiveness is limited in evaluating osseous changes, thereby necessitating its use, primarily when clinical and other methods suggest differential diagnoses, such as internally deranged TMJ or synovitis. Furthermore, recent advancements in imaging technologies, such as 3D printing and computer-aided design, facilitate precise surgical planning and simulations for patients with UCH. Because treatment depends on whether or not cartilage growth has ceased, functional imaging techniques, like bone scintigraphy through planar or tomographic images (SPECT), have been used for defining the metabolic condition of the affected condyle [8]. Bone scintigraphy uses phosphonate compounds labelled with technetium (99mTc), which exhibits a high affinity for osteoblastic activity that is increased in cases of active growth of the hyperplastic condyle. Different scintigraphic methods have been proposed for evaluating condylar hyperplasia, the relative activity between the two condyles being the most commonly used approach, with a sensitivity of 80–100% when the difference is greater than or equal to 10% [9]. Hodder et al. reported a normal relative uptake value range of 45–55% [10]; Ouyang et al. found 13% to be the upper normal relative uptake value [11]; Saridin et al. [12] concluded that for UCH patients, the analysis method of choice is the comparison of bone activity between the affected condyle and the contralateral condyle with AUC (0.93), yielding a sensitivity and specificity of 88%. Other authors have reported values by using ratios such as condyle counts/clivus counts: Fahey et al. reported normal values for ratios using the clivus as standard in 32 patients across three age ranges, and noted a trend toward lower values in older patients (0.73–1.88). They also found the normal right-to-left ratio to be in the range of 0.9–1.0 [13]. In a cohort of individuals with no condylar pathology, Anzola et al. reported normal reference values adjusted for age and sex using a condyle-by-condyle analysis, with activity ratios relative to the clivus uptake [14]. Emerging evidence underscores the potential benefits of hybrid SPECT/CT imaging in UCH diagnosis, which combines anatomical and metabolic data; however, these approaches have yet to demonstrate clear superiority over traditional SPECT techniques in delineating condylar metabolic activity [15,16]. Previous studies have highlighted specific CT findings indicative of active condylar growth, including significant mandibular deviation (greater than 6 mm) [17], increased anterior–posterior length of the mandible, and augmentation of ramus height [11]. The treatment of UCH aims to restore normal function and achieve a balanced profile, with SPECT bone scans guiding surgeons on the optimal operative technique. Treatment flowcharts highlight the significant role of SPECT, categorizing patients into two groups: a) those with active disease, who require condylectomy, and b) those with inactive disease, best treated with orthognathic surgery, genioplasty, or lower border reduction, among other techniques [17]. Despite the significance of SPECT in identifying actively growing hyperplastic condyles and its implications for therapeutic decision-making, it is essential to thoroughly evaluate the diagnostic accuracy of this method in assessing the metabolic condition of the affected condyle. Additionally, ongoing research into the pathophysiology of UCH and the development of more refined imaging techniques may provide further insights, potentially leading to improved diagnostic and treatment outcomes. The main objective of this study was to conduct a diagnostic test on a cohort of patients with confirmed condylar hyperplasia (CH) by performing a head-to-head comparison of two methods for assessing the metabolic activity of the condyles: a) the relative activity between the two condyles, and b) the individual quantitative assessment using the condyle/clivus ratio. This comparison aimed to facilitate the diagnostic approach for unilateral condylar hyperplasia (UCH) and bilateral condylar hyperplasia (BCH). The secondary objective was to describe the operational characteristics of each method and to analyze the trends in condyle/clivus ratios within the patient cohort. We hypothesized that the discriminative power, as assessed by the receiver operating characteristics (ROC) curve, is equivalent for both methods, the condyle/clivus ratio method and the relative uptake method, in contrast to the alternative hypothesis, which asserts that these powers are significantly different. By confirming this hypothesis, we aim to enhance the diagnostic strategies for CH, providing clearer guidance for clinical decision-making.

## 2. Materials and Methods

### 2.1. Study Design

This study involved a descriptive analysis of 139 individuals, including 91 patients with radiologically and clinically confirmed UCH or BCH, and 48 patients without the condition. An analytical cross-sectional study was conducted with retrospective data from diagnostic tests to compare the operational performance of the relative activity between the two mandibular condyles with individual quantitative assessments using the condyle/clivus ratio in patients with mandibular CH. The study recruited participants from the nuclear medicine department, from August 2019 to January 2024, with imaging evaluations conducted by nuclear medicine physicians. This research was approved by the hospital’s Ethics Committee (approval number CEIFUS 531-24). Due to the retrospective nature of the analysis, the requirement for written informed consent from the patients was waived.

### 2.2. Study Population

A non-probabilistic consecutive sampling method was performed; patients were evaluated according to the algorithm described by Nolte et al. [18]. Patients with CH were referred by maxillofacial surgeons to the nuclear medicine unit of our institution to assess condylar metabolic activity. All patients exhibited anamnestic or clinical presentations of progressive mandibular asymmetry, which were confirmed using computed tomography (CT). Confirmation of the patients’ clinical condition involved a combination of medical history, serial assessments of their clinical and radiological status, and clinical follow-ups, along with histopathological evaluations of the resected condyle, when feasible. All patients underwent ^99m^Tc HMDP SPECT bone scans. Image data sets from individuals without the condition were acquired from patients who were referred for bone scanning to evaluate suspected or known skeletal conditions unrelated to any temporomandibular pathology.

### 2.3. Bone SPECT Imaging

Patients were intravenously injected with ^99m^Tc-hydroxymethylene diphosphonate. The injected dose was calculated by adjusting the standard dose of 600 MBq based on the patient’s body surface area [19]. SPECT images were acquired using a dual-head gamma camera (Infinia, GE Healthcare, Waukesha, WI, USA) equipped with double-head, high-resolution, low-energy, parallel hole collimators. The SPECT study was performed using 120 projection images over a 360° rotation, with 20 s per projection in a 128 × 128 matrix. Transaxial, coronal and sagittal tomograms were reconstructed using a Butterworth filter and the ordered subset expectation maximization (OSEM) algorithm (two iterations, 10 subsets). To calculate relative activity (relative uptake), regions of interest (ROIs) of the same size were drawn over the heads of the two condyles. The left-to-right difference in activity between the condyles was calculated using the equation provided by Saridin et al.: relative activity = (counts per pixel in the ipsilateral ROI/counts per pixel in the ipsilateral + contralateral ROIs) ×100%. This calculation was performed using the maximum counts across all SPECT slices [20]. A relative uptake difference greater than 10% between the sides was considered indicative of active hyperplasia. For the individual quantitative analysis, the uptake ratios of the condyle to the clivus were calculated; the normal reference values provided by Anzola et al. [14] were utilized.

### 2.4. Statistical Analyses

A descriptive study involving 139 patients with 278 condyles was conducted. A comprehensive descriptive analysis of the variables of interest was performed to comply with the study objectives. The distribution of clinical and radiological characteristics was evaluated based on the presence or absence of the disease for each participant. Each condyle was considered as a sampling unit for the operational performance analysis. Measures of central tendency were employed to describe quantitative variables, while absolute and relative frequencies were used for qualitative variables. Central tendency measures were used to describe quantitative variables, and absolute and relative frequencies to describe qualitative variables. The sociodemographic characteristics of the population, along with the values of relative uptake and condyle/clivus ratios, were analyzed in a stratified manner for the UCH and BCH groups. The frequency of abnormal findings was calculated for the two methods: the relative uptake method (defined as a difference between sides greater than 10%) and the condyle/clivus ratio method (compared to the reference values reported by Anzola et al.). Furthermore, CT scan results were compared with the relative uptake and condyle/clivus ratios, stratified by type of hyperplasia. For individuals with positive findings on CT scans, the relative uptake values and ratios were described according to the type of hyperplasia. Operational characteristics were determined by calculating sensitivity, specificity, predictive values, likelihood ratios, and diagnostic odds ratios (DOR) for both the ratios and relative uptake. Additionally, receiver operating characteristic (ROC) curve values for both methods were computed and the curves were compared using the DeLong method. For the analysis of the strength of effect size, the assumptions of Cohen’s d were used: an effect size of d = 0.8 is considered strong, d = 0.5 is considered medium, and d = 0.2 is considered weak. A stratified analysis of the ratios and differences was performed based on the type of unilateral or bilateral CH contrasted with the CT scan results reported in participants with CH. The sample size calculation was performed using the pROC package in R version 4.01. The study was designed with a significance level (Type I error) of 0.05 and desired power of 0.95. Under the null hypothesis, the area under the curve (AUC) for the condyle/clivus ratio test was set at 0.80, while the AUC for the relative uptake was set at 0.60. An allocation ratio of 1 was utilized, and the sample size was adjusted to account for an expected loss rate of 10%, due to incomplete data. Consequently, a minimum sample size of 200 observational units was planned to ensure the study’s statistical validity. The analysis was conducted using Stata 17 MP software (Stata Corp, College Station, TX, USA).

## 3. Results

Data were collected from 139 participants, comprising 91 individuals with CH (mean age: 18 years; range: 15–23 years). Among these participants, 56 patients were categorized as having UCH, and 35 as having BCH; 48 individuals were included as a control group without the condition. The demographic characteristics of the patients are detailed in Table 1, according to the presence of the disease. The median age of participants was 20 years old (Interquartile range [IQR]: 16 years) and 54.68% of the population were females. A higher median age was observed in the control group, with 25% being 50 years or older.

Table 2 presents the values obtained for each of the evaluated methods across the two groups: patients with CH and those without the condition. The condyle/clivus ratio distinctly indicated elevated values in patients with CH, exceeding the normal cutoff adjusted for gender and age, with a median of 1.26 (range:1.00–1.66), compared to the normal value of ≤0.86.

Figure 1 illustrates the distribution of values for each method in both groups. Notably, the condyle/clivus ratio was consistently elevated with values above the cutoff in patients with CH, while the relative uptake exhibited normal values in a considerable number of individuals from both groups.

Table 3 outlines the findings related to the operational performance of each method. The condyle/clivus ratio method demonstrated superior sensitivity values (90.1% compared to 40.7% for the relative uptake method), as well as a more favorable negative likelihood ratio (LR-) of 0.13 versus 0.82.

The ROC curve was calculated for the tests being compared, revealing a greater area under the curve (AUC) for the classification based on ratios. Specifically, the condyle/clivus ratio yielded an AUC of 0.8360 (95% CI: 0.7884–0.8840), indicating a strong effect size of 0.8. In contrast, the relative uptake method yielded an AUC of 0.5679 (95% CI: 0.5106–0.6251), corresponding to a weak effect size of 0.2. Statistically significant differences were observed for the two curves (*p* value for chi square = 0.0001) (Figure 2).

Table 4 presents the discriminatory capacity of both tests within the CH population; results are categorized according to CT findings and the type of condylar hyperplasia (UCH and BCH). Among patients with UCH, the relative uptake method accurately identified 66% of those with CH, but missed 33.93%. In contrast, the condyle/clivus ratio method correctly identified 98.21% of the condyles that were positive on CT; however, for the normal condyles (negative CT result), 69.64% exhibited ratios above the normal threshold. For patients with BCH, the relative uptake method did not identify any cases, whereas the condyle/clivus ratio method successfully identified 100% of the affected condyles.

Figure 3 presents a SPECT bone scan image of a patient with left UCH. The condyle/clivus ratios exceed the normal range, indicating an active left UCH associated with biomechanical stress on the right condyle.

## 4. Discussion

This analytical cross-sectional study evaluated differential diagnostic approaches for active CH using bone SPECT techniques. It compared the effectiveness of two methods: relative activity assessment between the two condyles and quantitative analysis using the condyle/clivus ratio. The condyle/clivus ratio method exhibited superior diagnostic performance: sensitivity 90.1% (95% CI:84.1–94) vs. 40.7% (95% CI:33.5–48.2), specificity 77.1% (95% CI:67.4–85) vs. 72.9% (95% CI:62.9–81.5), and a more favorable negative likelihood ratio (0.13 vs. 0.82). Additionally, it demonstrated a higher area under the curve (AUC) of 0.8360 compared to 0.5679 for relative uptake, with statistically significant differences (*p* = 0.0001). The condyle/clivus ratio method also outperformed the relative uptake method in identifying affected condyles in both unilateral and bilateral CH. Condylar hyperplasia, characterized by significant facial asymmetry, is more common in women and younger individuals, with unilateral condylar hyperplasia (UCH) being more prevalent than bilateral hyperplasia [21]. The results observed in this cohort closely align with these established trends (Table 1). The treatment approach depends on ongoing condylar growth, making accurate assessments of metabolic activity essential for guiding surgical interventions. Due to the limitations of computed tomography (CT) in this regard, single-photon emission computed tomography (SPECT) is a valuable complementary imaging technique for identifying actively growing hyperplastic condyles [8]. SPECT and SPECT/CT are preferred diagnostic tools for assessing the metabolic condition of the condyle [22], with a diagnostic threshold of a 10% difference in relative uptake recommended for identifying active UCH. However, discrepancies in threshold values due to assessment methods and ethnic variations exist. In patients with BCH, the side-to-side difference often falls below 10%, limiting the clinician’s ability to accurately detect abnormalities. This compromise in diagnostic accuracy raises the risk of misclassification, highlighting the need for refined diagnostic criteria tailored to specific clinical scenarios. The results of this study show a significant difference in condyle/clivus ratios between healthy individuals and those with pathology, highlighting its distinct advantage over the relative uptake method (Table 2, Figure 1). The diagnostic performance findings indicate that the condyle/clivus ratio is significantly more sensitive than the relative uptake method, making it a more valuable tool for disease detection. Additionally, the low negative likelihood ratio (LR-) for the condyle/clivus ratio enhances its utility in ruling out the disease, while the relative uptake method has limitations that could adversely affect diagnosis and patient management (Table 3). A comparison of the two methods reveals better diagnostic performance for the condyle/clivus ratio, evidenced by an AUC of 0.8360 (Figure 2), and a statistically significant difference (*p* = 0.000) compared to the relative uptake method. This finding suggests that the condyle/clivus ratio may be a more reliable tool for diagnosing condylar hyperplasia, warranting its prioritization by clinicians for more accurate diagnoses. If demonstrated consistently across diverse patient populations, this method could standardize diagnostic approaches in clinical settings, enhancing patient management and outcomes. Notably, the sensitivity and specificity values reported in this study are not directly comparable to those in the literature, as those were based solely on UCH patients, while this study also included patients with BCH. Frequencies observed for each method were described in patients categorized as UCH or BCH (Table 4). Among 56 UCH patients, the relative uptake method correctly identified 37 patients (66%) but misclassified 19 patients (33.93%) as normal, masking the underlying pathology with a relative uptake of under 10%. This finding underscores the challenges of interpreting relative differences in UCH patients and the limitations of relying solely on established relative uptake thresholds. In contrast, the condyle/clivus ratio accurately classified 55 of 56 condyles (98.21%) as hyperplastic on CT, but detected increased abnormal metabolism in 39 of 56 condyles (69.64%) that were reported as normal on CT. In patients with BCH, only the condyle/clivus ratio detected all hyperplastic condyles, while the relative uptake method failed to identify any abnormal condyles, since all recorded indices were below 10%. Given that bone scintigraphy can detect metabolic changes from mechanical stress [23], it is reasonable to hypothesize that the contralateral condyle of the one affected by active UCH may also experience biomechanical stress, resulting in increased tracer uptake. This phenomenon highlights the interconnected nature of biomechanical stress in joint compartments and underscores the need to consider the broader physiological context when interpreting scintigraphic findings. Thus, clinicians should be cautious and maximize clinical judgment when evaluating radionuclide images to ensure accurate diagnosis and effective patient management. The unique biomechanical characteristics of the temporomandibular joint (TMJ) play a significant role in understanding the scintigraphic findings in patients with UCH. The TMJ experiences substantial loading during occlusion, with unilateral occlusion leading to increased stress on the non-working contralateral TMJ [24]. This load-bearing function is essential for maintaining equilibrium during masticatory function, making the TMJ one of the most heavily loaded joints per unit area in the body [25]. These findings underscore the utility of the condyle/clivus ratio method in assessing metabolic activity and identifying biomechanical stress conditions associated with CH, highlighting the importance of individualized diagnostic approaches and the complementary role of bone scintigraphy in evaluating complex skeletal pathologies. Several studies have demonstrated the ability to assess active CH using various techniques, such as the correlation of mandibular deviation magnitude in CT with SPECT uptake [26], as well as the quantitative analysis of SPECT/CT using standardized uptake values (SUVs) [27]. Additionally, novel technologies, such as ^18^F-fluoride PET-CT [28], offer improved resolution and allow for the evaluation of metabolic activity of the condyle in vivo [29,30]. Some studies also show the potential utility of artificial intelligence [25,31]. These advancements highlight the evolving landscape of imaging modalities and their role in refining the diagnosis and management of CH. Although this study provided valuable insights into the diagnostic approaches for active CH, some limitations must be considered. First, while the study included a substantial number of patients, the sample size may still limit the generalizability of the findings. Larger multicenter studies could provide more robust conclusions regarding the diagnostic efficacy of the condyle/clivus ratio and relative methods across diverse populations. Second, the retrospective nature of the study may have introduced selection bias, as data were collected from previously evaluated patients. Third, conducting the research at a single institution may potentially impact the applicability of the results in other clinical settings. Fourth, the control group comprised 48 individuals without CH, but the lack of further stratification within this group for other potential confounding conditions may limit the interpretability of the results. Additionally, while the condyle/clivus ratio showed superior performance, the clinical significance of identifying patients with subtle differences in uptake remains to be fully elucidated. Further investigations are needed to determine how these diagnostic methods influence treatment decisions and outcomes. This study’s findings emphasize critical considerations in determining condylar growth activity in patients with CH. It highlights the importance of precisely evaluating the patient, preferably through individual analysis of each condyle, to accurately determine its metabolic activity. The findings also emphasize that nuclear medicine laboratories should validate both their reference values and their measurement methods to ensure that they are suitably accurate and reproducible. Additionally, structural evaluations using CT should be mandatory to aid in interpreting abnormalities detected on SPECT, while mitigating the false positive results inherent to this technique. Notably, a mechanical overload in the contralateral condyle may confound the results, particularly when relative analysis methods are used. These considerations are pivotal for refining diagnostic approaches and enhancing the clinical management of condylar hyperplasia.

## 5. Conclusions

This study demonstrates that the condyle/clivus ratio method significantly outperforms the relative uptake method in diagnosing active CH, and highlights the importance of incorporating this method into clinical practice for more accurate identification of affected condyles in both unilateral and bilateral cases. It is crucial for laboratories to validate their normal reference values to ensure consistent and reliable diagnostic interpretations tailored to their specific populations.

## Figures and Tables

**Figure 1 diagnostics-14-02548-f001:**
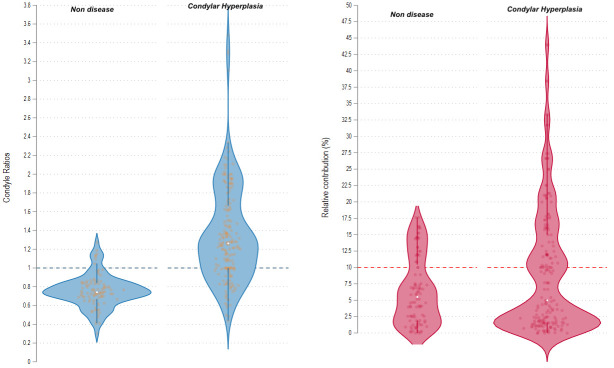
Graphical representation of the distribution of the values obtained under the two methods in individuals with and without the condition.

**Figure 2 diagnostics-14-02548-f002:**
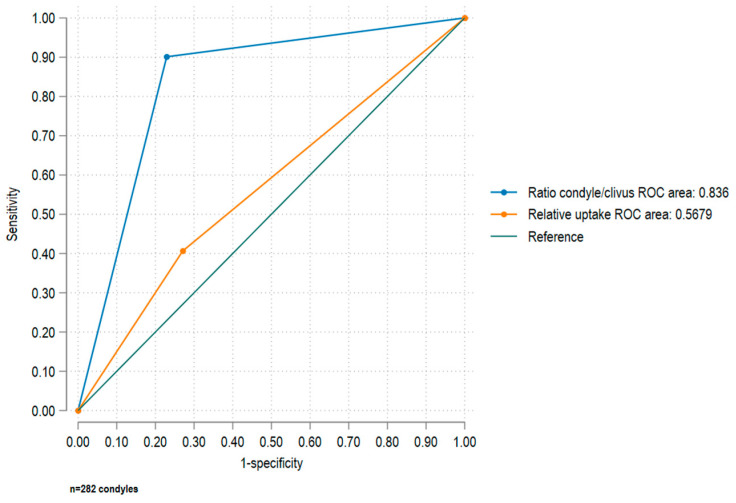
ROC Analysis.

**Figure 3 diagnostics-14-02548-f003:**
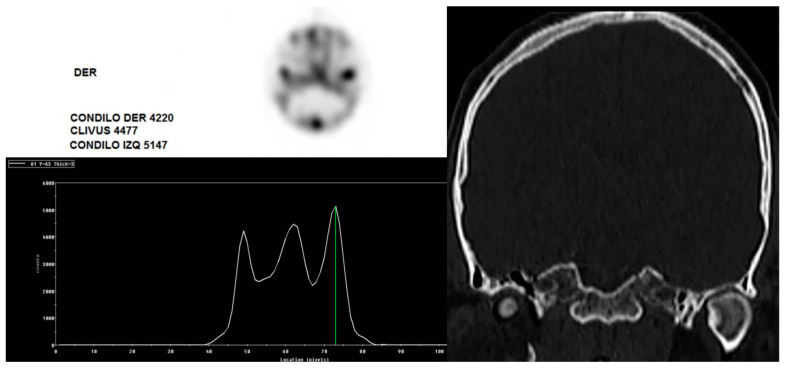
Left image shows bone scan SPECT with high uptake of the radiotracer in both condyles, with condyle/clivus ratio above normal values. Right image confirm on CT scan left hyperplasia with normal right condyle.

**Table 1 diagnostics-14-02548-t001:** Demographic characteristics of the patients.

Variables	Non Disease	Condylar Hyperplasia	Total
	N = 48	N = 91	N = 139
Sex			
Female	26 (54.17%)	50 (54.95%)	76 (54.68%)
Male	22 (45.83%)	41 (45.05%)	63 (45.32%)
Age (years)	33.50 (22.00–49.50)	18.00 (15.00–23.00)	20.00 (16.00–32.00)
Age group			
10 or younger	9 (18.75%)	58 (63.74%)	67 (48.20%)
10–20 years old	9 (18.75%)	26 (28.57%)	35 (25.18%)
20–30 years old	11 (22.92%)	5 (5.49%)	16 (11.51%)
30–40 years old	7 (14.58%)	1 (1.10%)	8 (5.76%)
50 or older	12 (25.00%)	1 (1.10%)	13 (9.35%)
Clinical status			
Bilateral Hyperplasia	0 (0.00%)	35 (38.46%)	35 (25.18%)
Unilatera Hyperplasial	0 (0.00%)	56 (61.54%)	56 (40.29%)
Non disease	48 (100.00%)	0 (0.00%)	48 (34.53%)

Data are presented as median (IQR) for continuous measures, and n (%) for categorical measures.

**Table 2 diagnostics-14-02548-t002:** Values for each method in patients with and without the condition.

Variables	Non Disease	Condylar Hyperplasia	Total
	N = 96	N = 182	N = 278
Ratio condyle/clivus	0.74 (0.67–0.84)	1.26 (1.00–1.66)	1.00 (0.77–1.36)
Relative uptake	0.06 (0.02–0.10)	0.05 (0.02–0.15)	0.05 (0.02–0.12)

Data are presented as median (IQR) for continuous measures, and n (%) for categorical measures.

**Table 3 diagnostics-14-02548-t003:** Operational characteristics for relative uptake and condyle/clivus ratio methods.

**Relative Uptake**			
**Variables**	**Non Disease**	**Condylar Hyperplasia**	**Total**
	**N = 96**	**N = 182**	**N = 278**
Relative uptake			
dif < 10% (normal)	70 (72.92%)	108 (59.34%)	178 (64.03%)
dif ≥ 10% (abnormal)	26 (27.08%)	74 (40.66%)	100 (35.97%)
**Results**	**Estimate**	**95% CI**	
Sensitivity	40.7%	(33.5–48.2)	
Specificity	72.9%	(62.9–81.5)	
ROC area	0.57	(0.51–0.63)	
Likelihood ratio (+)	1.49	(1.03–2.15)	
Likelihood ratio (−)	0.82	(0.69–0.97)	
Diagnostic Odds ratio	1.83	(1.07–3.12)	
VPP	74	(64.3–82.3)	
VPN	39.3	(32.1–46.9)	
**Ratio Condyle/Clivus**			
**Variables**	**Non Disease**	**Condylar Hyperplasia**	**Total**
	**N = 96**	**N = 182**	**N = 278**
Ratio condyle/clivus			
ratio < cut off (normal)	74 (77.08%)	18 (9.89%)	92 (33.09%)
ratio > cut off (abnormal)	22 (22.92%)	164 (90.11%)	186 (66.91%)
**Results**	**Estimate**	**95% CI**	
Sensitivity	90.1%	(84.1–94)	
Specificity	77.1%	(67.4–85)	
ROC area	0.84	(0.79–0.88)	
Likelihood ratio (+)	3.88	(2.69–5.58)	
Likelihood ratio (−)	0.13	(0.08–0.20)	
Diagnostic Odds ratio	29.4	(15–57.7)	
VPP	88.2	(82.6–92.4)	
VPN	80.4	(70.9–88)	

**Table 4 diagnostics-14-02548-t004:** Discriminatory capacity for each method according to CT results and type of CH.

	**Unilateral Hyperplasia**		
**Variables**	**Negative CT Result**	**Positive CT Result**	**Total**
Relative uptake	0.12 (0.10–0.18)	0.10 (0.04–0.20)	0.12 (0.07–0.19)
Relative uptake			
dif < 10% (normal)	6 (24)	13 (41.94)	19 (33.93)
dif ≥ 10% (anormal)	19 (76)	18 (58.06)	37 (66.07)
Ratio condyle/clivus ^a^	0.97 (0.77–1.10)	1.19 (1.00–1.40)	1.04 (0.85–1.27)
Ratio condyle/clivus ^a^			
ratio < cut off (normal)	17 (30.36)	1 (1.79%)	18 (16.07%)
ratio > cut off (abnormal)	39 (69.64)	55 (98.21%)	94 (83.93%)
	**Bilateral Hyperplasia**		
**Variables**	**Negative CT Result**	**Positive CT Result**	**Total**
Relative uptake	--	0.01 (0.01–0.02)	0.01 (0.01–0.02)
Relative uptake			
dif < 10% (normal)	--	35 (100.00%)	35 (100.00%)
dif ≥ 10% (abnormal)	--	0 (0.00%)	0 (0.00%)
Ratio condyle/clivus ^b^	--	1.72 (1.36–1.98)	1.72 (1.36–1.98)
Ratio condyle/clivus ^b^			
ratio < cut off (normal)	--	0 (0.00%)	0 (0.00%)
ratio > cut off (abnormal)	--	70 (100.00%)	70 (100.00%)

Data are presented as median (IQR) for continuous measures, and n (%) for categorical measures. ^a^ n ratio = 112; ^b^ n ratio = 70.

## Data Availability

The original contributions presented in the study are included in the article, further inquiries can be directed to the corresponding author.

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
