# Peer review of "Diagnostic Efficacy of Bone SPECT Techniques in Differentiating Unilateral and Bilateral Condylar Hyperplasia"

_diagnostics, 2024, doi:10.3390/diagnostics14222548_

Round 1
Reviewer 1 Report
Comments and Suggestions for Authors
This study employs an analytical cross-sectional design to systematically evaluate the effectiveness of bone SPECT techniques in diagnosing unilateral and bilateral condylar hyperplasia by comparing relative activity assessment between condyles and quantitative analysis via the condyle/clivus ratio. The study methods are scientifically rigorous, and the data analysis is comprehensive, providing strong support for the reliability of the results. This manuscript provides new insights and evidence for the application of bone SPECT techniques in diagnosing condylar hyperplasia, particularly highlighting the superiority of the condyle/clivus ratio method. The manuscript not only enriches academic knowledge in the field but also offers clinicians a more reliable diagnostic tool, with significant potential for clinical translation. In future studies are recommended to expand the sample size and validate the findings in different populations to ensure broader applicability. Additionally, exploring other potential quantitative analysis metrics could further enhance the diagnostic value of bone SPECT techniques in condylar hyperplasia.
Author Response
Dear reviewer, Thank you for your valuable feedback and for taking the time to review our manuscript titled “ Diagnostic Efficacy of Bone SPECT Techniques in Differentiating Unilateral and Bilateral Condylar Hyperplasia” We appreciate your constructive comments, which have helped us enhance the clarity and quality of our work. Below, we outline our responses to your minor corrections regarding to the improvement of the introduction section, and the corresponding changes made in the manuscript. All changes in the manuscript are highlighted for your convenience. We believe that the revisions have strengthened the paper, and we hope it meets your approval. Thank you once again for your insights and support:
Answer
Unilateral condylar hyperplasia (UCH) is a pathological condition of the condylar cartilage of the mandible, caused by a somatic mutation resulting in excessive effects of growth hormones (growth factors like IGF-1), which promotes temporo-mandibular cartilage overgrowth by enhancing chondrocytes proliferation via mitogen-activated protein kinase/extracellular signal-regulated kinase (MAPAK-ERK) pathway [1,2,3] leading to pronounced facial asymmetry and functional impairment, such as mal occlusion and altered temporomandibular joint (TMJ) function. Four observations have supported the hypothesis that UCH is a primarily disorder of the condylar cartilage: a) The deformity of the condylar process is more extensive than that of other parts of the mandible; b) when the overgrowth is present, the bone scan shows localized abnormal uptake in the condyle; c) resection of the condylar cartilage suppresses the abnormal growth; and d) histopathologic changes are limited to the condylar cartilage and the adjacent subchondral bone [4].Furthermore, genetic factors may predispose individual to this condition, emphasizing the need for comprehensive genetic studies to uncover potential biomarkers. Most patients present at adolescence or young adulthood [5] mostly seen between 11 and 30 years of age, males and females being equally affected;early diagnosis and intervention are critical to prevent complications, including temporomandibular disorders, persistent pain, and permanent skeletal deformities.Clinical evaluations, combined with advanced imaging techniques, play a pivotal role in the diagnosis and treatment planning for UCH. Computed tomography (CT) is recognized as the standard imaging modality, providing detailed anatomical and morphological information essential for distinguishing condylar hyperplasia from other conditions, such as osteochondroma and malignant tumors [6]. For diagnostic purposes, cone-beam CT is often preferred due to its superior resolution and reduced radiation exposure, making it particularly useful in young patients [7]. While magnetic resonance imaging (MRI) provides excellent soft tissue characterization, its effectiveness is limited in evaluating osseous changes, thereby necessitating its use primarily when clinical and other methods suggest differential diagnoses, such as internally deranged TMJ or synovitis. Furthermore, recent advancements in imaging technologies, such as 3D printing and computer-aided design, facilitate precise surgical planning and simulations for patients with UCH. Because treatment depends on whether or not cartilage growth has ceased, functional imaging techniques like bone scintigraphy through planar or tomographic images (SPECT) have been used for defining the metabolic condition of the affected condyle [8]. Bone scintigraphy uses phosphonate compounds labelled with technetium (99mTc), which exhibit high affinity for osteoblastic activity, which is increased in cases of active growth of the hyperplasic condyle. Different scintigraphic methods have been proposed for evaluating condylar hyperplasia, being the relative activity between the two condyles the most commonly used approach, with a sensitivity of 80-100% when the difference is greater than or equal to 10% [9]. Hodder et al. reported a normal relative uptake value range of 45%-55% [10]; Ouyang et al found 13% to be the upper normal relative uptake value [11], Saridin et al [12] concluded that for UCH patients, the analysis method of choice is the comparison of bone activity between the affected condyle and the contralateral with AUC (0.93), yielding a sensitivity and specificity of 88%. Other authors have reported values by using ratios such as condyle counts/clivus counts: Fahey et al reported in 32 patients, normal values for ratios using the clivus as standard in three age ranges, and noticed a trend to lower values in older patients (0.73-1.88); they found the normal right-to-left ratio to be in the range of 0.9-1.0 [13]. In a cohort of individuals with no condylar pathology, Anzola et al reported normal reference values adjusted for age and sex through a condyle-by-condyle analysis, by using activity ratios against the clivus uptake [14]. Emerging evidence underscores the potential benefits of hybrid SPECT/CT imaging in UCH diagnosis, combining anatomical and metabolic data; however, these approaches have yet to demonstrate clear superiority over traditional SPECT techniques in delineating condylar metabolic activity [15,16]. Previous studies have highlighted specific CT findings indicative of active condylar growth, including significant mandibular deviation (greater than 6mm) [17], increased anterior-posterior length of the mandible, and an augmentation in ramus height [11]. The treatment of UCH aims to restore normal function and achieve a balanced profile, with SPECT bone scans guiding surgeons on the optimal operative technique. Treatment flowcharts highlight the significant role of SPECT, categorizing patients into two groups: a) those with active disease who require condilectomy, and b) those with inactive disease best treated with orthognathic surgery, genioplasty, or lower border reduction, among other techniques [17]. Despite the significance of SPECT in identifying actively growing hyperplastic condyles and its implications for therapeutic decision-making, it is essential to thoroughly evaluate the diagnostic accuracy of this method in assessing the metabolic condition of the affected condyle. Additionally, ongoing research into the pathophysiology of UCH and the development of more refined imaging techniques may provide further insights, potentially leading to improved diagnostic and treatment outcomes. The main objective of this study was to conduct a diagnostic test study in a cohort of patients with confirmed condylar hyperplasia (CH), by performing a head-to-head comparison of two methods for assessing the metabolic activity of the condyles: a) the relative activity between the two condyles, and b) the individual quantitative assessment through the condyle/clivus ratio. This comparison aimed to facilitate the diagnostic approach for unilateral condylar hyperplasia (UCH) and bilateral condylar hyperplasia (BCH). The secondary objective was to describe the operational characteristics of each method and to analyze the trends in condyle/clivus ratios within the patient cohort.We hypothesize that the discriminative power, as assessed by the receiver operating characteristics (ROC) curve, is equivalent for both methods: the condyle/clivus ratio and relative uptake, in contrast to the alternative hypothesis, which asserts that these powers are significantly different. By confirming this hypothesis, we aim to enhance the diagnostic strategies for CH, providing clearer guidance for clinical decision-making.
Reviewer 2 Report
Comments and Suggestions for Authors
After reviewing the work, I have the following comments. In particular, please add effect sizes.
- 'overgrowth by enhancing chondrocytes proliferation via MAPK-ERK pathway [1,2,3] leading' – please standardize the font throughout the text.
- '[4]. Most' – add the missing period.
- '(OR:3.51, CI: 95%: 1.27-9.72)' – I suggest not including statistical results in the introduction.
- Line 76 and line 82 – should these be new paragraphs?
- Line 94 – please state the research hypotheses.
- Line 120 – should this be a new paragraph?
- Which software was used to calculate the sample size?
- 'power of 0.95' – can the authors justify this power level?
- 'The sociodem-' – add a paragraph break. I will not mark this type of error again.
- 'Table 3. Operational' – in my version, this table appears cut off halfway; please correct the table insertion, as I am unable to verify the results.
- Please add effect size analyses to the statistical analysis and overall results.
- Please move Table 4 to the Results section.
- In the references, bold the publication year to meet journal requirements.
Best regards.
Author Response
Dear reviewer, Thank you for your valuable feedback and for taking the time to review our manuscript titled “ Diagnostic Efficacy of Bone SPECT Techniques in Differentiating Unilateral and Bilateral Condylar Hyperplasia” We appreciate your constructive comments, which have helped us enhance the clarity and quality of our work. Below, we outline our responses to your minor corrections and the corresponding changes made in the manuscript. All changes in the manuscript are highlighted for your convenience. We believe that the revisions have strengthened the paper, and we hope it meets your approval. Thank you once again for your insights and support.
Answers:
After reviewing the work, I have the following comments. In particular, please add effect sizes.
- 'overgrowth by enhancing chondrocytes proliferation via MAPK-ERK pathway [1,2,3] leading' – please standardize the font throughout the text.
A: it was corrected in the text according to your observation:
which promotes temporo-mandibular cartilage overgrowth by enhancing chondrocytes proliferation via mitogen-activated protein kinase/extracellular signal-regulated kinase (MAPAK-ERK) pathway [1,2,3] leading to pronounced facial asymmetry and functional impairment, such as mal occlusion and altered temporomandibular joint (TMJ) function. [lines 34-35]
- '[4]. Most' – add the missing period
It was corrected.
- '(OR:3.51, CI: 95%: 1.27-9.72)' – I suggest not including statistical results in the introduction.
It was corrected.
- Line 76 and line 82 – should these be new paragraphs.
It was checked and corrected.
- Line 94 – please state the research hypotheses.
It was added to the last paragraph according to your suggestion
We hypothesize that the discriminative power, as assessed by the receiver operating
characteristics (ROC) curve, is equivalent for both methods: the condyle/clivus ratio and
relative uptake, in contrast to the alternative hypothesis, which asserts that these powers are
significantly different. By confirming this hypothesis, we aim to enhance the diagnostic
strategies for CH, providing clearer guidance for clinical decision-making. [lines 106-110]
- Line 120 – should this be a new paragraph?
It was checked and corrected
- Which software was used to calculate the sample size?
It was included in statistics section
The sample size calculation was performed using the pROC package in R version 4.01. T [lines 186-187]
- 'power of 0.95' – can the authors justify this power level?
It was included and explained in the statistics section
The study was designed with a significance level (Type I error) of 0.05 and desired power of 0.95. Under the null hypothesis, the area under the curve (AUC) for the condyle/clivus ratio test was set at 0.80, while the AUC for the relative uptake was set at 0.60. An allocation ratio of 1 was utilized, and the sample size was adjusted to account for an expected loss rate of 10% due to incomplete data. Consequently, a minimum sample size of 200 observational units was planned to ensure the study´s statistical validity. [lines 187-192]
- 'The sociodem-' – add a paragraph break. I will not mark this type of error again.
It was checked and corrected
- 'Table 3. Operational' – in my version, this table appears cut off halfway; please correct the table insertion, as I am unable to verify the results.
It was corrected and made to match the template format. You can check the table in the attached file below.
- Please add effect size analyses to the statistical analysis and overall results.
It was included in the statistics and results sections according to your recommendation
For the analysis of the strength of effect size, the assumptions of Cohen’s d were used: an effect size of d=0.8 is considered strong, d=0.5 is considered medium, and d= 0.2 is considered weak. [lines 181-183]
The ROC curve was calculated for the tests being compared, revealing a greater area under the curve (AUC) for the classification based on ratios. Specifically, the condyle/clivus ratio yielded an AUC of 0.8360 (95% CI: 0.7884-0.8840), indicating a strong effect size of 0.8. In contrast, the relative uptake method yielded an AUC of 0.5679 (95% CI: 0.5106-0.6251), corresponding to a weak effect size of 0.2. Statistically significant differences were observed for the two the curves (p value for chi square=0.0001) [lines 241-246]
- Please move Table 4 to the Results section.
Checked and corrected.
